# OpenReview forum: "Moral High Ground: A text-based games benchmark for moral evaluation"
_ICLR.cc/2024/Conference — ICLR 2024 Conference Withdrawn Submission_

### Official Review · Reviewer_6LQq · 2023-10-13

**Soundness:** 1 poor
**Presentation:** 2 fair
**Contribution:** 1 poor
**Rating:** 1
**Confidence:** 4

**Summary:**

This work seeks to establish a benchmark to establish the morality and business ethics of large language models. They construct their benchmark from two sources: IBM's Business Conduct Guidelines and the Social Chemistry dataset. From these sources, they generate extensive-from games that embody different morals/ethics. They then evaluate a suite of LLMs on their ability to play these games. They additionally showed that a small-LLM fine-tuned on their dataset is able to achieve similar performance as a much larger LLM.

**Strengths:**

- Investigates an important problem of measuring the value alignment of LLMs.

**Weaknesses:**

- Morality and ethics are not well defined in general nor in the context of this paper. This is largely because they are often contextually defined.
- It's not clear how the concepts sourced to generate games should constitute a benchmark and morality and ethics.
  - The BCG is just a handbook for IBM. Why is this one handbook the choice? What were the other options? The sections of these things seem really abstract.
  - Social Chemistry appears to not actually be a benchmark or definition of morality, but instead about compatibility between two agents.
- How the games are constructed from the source datasets is not described, only the mechanics of playing a game.
- There's no analysis of the constructed games, only the empirical performance of some LLMs. Recommend following at least a framework like "Datasheets for Datasets".

**Questions:**

- Sec 3.1, if the purpose of this work is to build a benchmark why are you filtering down the data?
  - Why choose to filter specifically the harder examples that might provide better insights into the quality of the models?
  - Could the authors provide more details as to how they deemed a question was too "abstract" and must be filtered? This seems like a point of failure, because authors opinions could overly influence the metrics.
  - Please provide further details about the number of questions considered, the total number of final questions, and information about the set of withheld question and why they were withheld.
  - How were the stories created that realized these different criteria? Why only one story?
- Sec 3.1 and 3.2 are missing basic analysis into the input data and the constructed games.
- It appears the aliases that were hand-constructed were essential for measuring the quality of current models. This unfairly biases the games to methods that were treated with this extra effort. Therefore, aliases must be further constructed for all future methods to ensure a more fair comparison --- where fair here is hard to quantify or ensure.
  - For example, consider a new LLM that uses broken English, but has a high morality/ethical position. This method may perform disproportionally poorly.
- Your results on the Flan models suggest a simple confounder for drawing any results of this dataset: it could simply measure a model's ability to follow instructions rather than its morality. I was hoping the authors might be able to speak more about this.

**Nit**
- There are issues with using `\cite`, `\citet`, and `\citep`.
- It seems like the font was modified.
- Sec 1, Par 2, "jailbroken" not defined.

**Details Of Ethics Concerns:**

This paper attempts to define benchmarks for concepts such as "morality" and "ethics" to evaluate LLMs. Any issues in these definitions can misalign the incentives of these models resulting in discrimination, bias, and fairness concerns that can potentially harm different groups. This paper does not treat the severity of these definitions with the rigour and care that would be needed to mitigate potential negative outcomes.

---

> ### Author Response · Authors · 2023-11-16
>
> * Clarification: social chemistry is indeed not a comprehensive benchmark for morality, but it does offer a useful starting point due to its clearly categorized and scored samples of situations and actions.  These cover a wide variety of individual scenarios across a more limited moral scope.
>
> * Section 3.2.2: added clarification on how games were created from the social chemistry dataset.
>
>   * “These stories are not necessarily based on the situations within this dataset; they are created manually by using the axes as inspiration for the story’s topic.“
>
>   * Our strategy was similar for BCG games.  We did not follow a precise method for creating games.  We primarily focused on a variety of relevant topics and created each scenario from one starting moral principle.
>
> * Section 3.2.1 (review refers to this as 3.1): reworded the sentence on filtering out some BCG topics to better reflect our intentions.
>
>   * “In our analysis, we looked for subsections of the BCG that focused on simple, grounded principles. Examples of these are "avoid corrupt practices or bribery" or "be honest". We focused on sections that were generalizable to most business environments, avoiding IBM-specific policies. In-game conversations are based off of the topics in these sections and are not taken directly from the document.“
>
> * Clarification: this is a very good point.  While the Flan models are not the best at following instructions, they still demonstrate some directed moral reasoning.  Even as different orders of valid actions are presented, it still follows a series of similar paths on each game.  Although its moral reasoning is not the best on all metrics, it does consistently choose from a subset of actions rather than completely random instruction following.
>
> * Section 1: better defined what we consider a jailbreak.
>
>   * “These jailbreaks, exploits that can cause a model to ignore safety training and produce immoral/unethical output, represent a severe and tangible threat, even to those that do not directly interact with their models.“
>
> * Changed \cite to \citep or \citet where appropriate.

---

> > ### Comment · Reviewer_6LQq · 2023-11-20
> >
> > Thank you for answering some of my questions and concerns. Unfortunately, the majority of major concerns still remain unresolved. Therefore, I will keep my original evaluation of the manuscript.

---

### Official Review · Reviewer_T1SJ · 2023-10-27

**Soundness:** 2 fair
**Presentation:** 3 good
**Contribution:** 2 fair
**Rating:** 5
**Confidence:** 4

**Summary:**

The objective of this paper is to evaluate the moral reasoning abilities of large
language models using text-based games. To begin, the authors develop these
text-based games in a conversational manner and generate training datasets in
the form of triplets comprising prompts, responses, and scores.

**Strengths:**

- The topic of ethics and large language models is becoming increasingly
important and urgent to address.
- The idea of discovering moral reasoning through conversation makes sense.

**Weaknesses:**

While the primary goal of this paper is
to introduce a new benchmark, the proposed benchmark does not sufficiently
demonstrate its potential for widespread use.

- The paper mentions that the game scores are manually annotated, but
it lacks an introduction to any specifics regarding human annotation or
evaluations.
- The sources of the games are derived from ’Business Conduct Guidelines’
and ’Social Chemistry,’ which may not be sufficient sources for defining
general ’moral’ or ethical reasoning, limiting the broader applicability of
this benchmark.
- There are several related works that discuss moral annotation [ZYW + 22,
JHB + 21], and benchmarks consider moral behavior in text-based games
with large language models [PCZ + 23]. However, this paper does not discuss these specific papers.


[JHB + 21] Liwei Jiang, Jena D Hwang, Chandra Bhagavatula, Ronan Le Bras,
Maxwell Forbes, Jon Borchardt, Jenny Liang, Oren Etzioni,
Maarten Sap, and Yejin Choi. Delphi: Towards machine ethics and
norms. arXiv preprint arXiv:2110.07574, 6, 2021.

[PCZ + 23] Alexander Pan, Jun Shern Chan, Andy Zou, Nathaniel Li, Steven
Basart, Thomas Woodside, Hanlin Zhang, Scott Emmons, and Dan
Hendrycks. Do the rewards justify the means? measuring trade-offs
between rewards and ethical behavior in the machiavelli benchmark.
In International Conference on Machine Learning, pages 26837–
26867. PMLR, 2023.

[ZYW + 22] Caleb Ziems, Jane Yu, Yi-Chia Wang, Alon Halevy, and Diyi Yang.
The moral integrity corpus: A benchmark for ethical dialogue sys-
tems. In Proceedings of the 60th Annual Meeting of the Associa-
tion for Computational Linguistics (Volume 1: Long Papers), pages
3755–3773, Dublin, Ireland, May 2022. Association for Computa-
tional Linguistics.

**Questions:**

- How do you position your work in relation to the above mentioned related work?
- How do you justify using 'Business Conduct Guidelines' as a basis for moral reasoning?

---

> ### Author Response · Authors · 2023-11-16
>
> * Section 5.1: added more information about who the annotators were and measures to reduce bias.
>
>   * “These games were created and annotated manually by the authors with response scores.“
>
>   * “Discussion between authors was held on scores to avoid significant bias.“
>
> * Section 2: reworked the related works section to include more relevant papers and references such as [JHB + 21], [ZYW + 22], and others that were not listed in the review.
>
> * Section 3.2: added a brief justification for including BCG document principles in our games.
>
>   * “We chose this document for its applicability to general business environments. Other moral corpora lack these business ethics principles.“
>
>   * “We focused on sections that were generalizable to most business environments, avoiding IBM-specific policies.“

---

> > ### Comment · Reviewer_T1SJ · 2023-11-22
> >
> > Thank you for the reply and incorporating some of the comments. However, after reading also the other reviewers comments, I think there are too many remaining issues and will keep my rating.

---

### Official Review · Reviewer_N9dw · 2023-11-02

**Soundness:** 3 good
**Presentation:** 3 good
**Contribution:** 3 good
**Rating:** 5
**Confidence:** 3

**Summary:**

This paper introduces a benchmark featuring a series of text-based games to evaluate the moral reasoning of language models. These games engage the model through multi-turn dialogues. Within a given dialogue context, the model can select from multiple valid actions, each associated with a distinct moral score set manually.
The author tested serval leading open-source language models on this benchmark. Among them, Flan-T5 was fine-tuned and its performance was assessed on both the proposed benchmark and other moral corpora. Results suggest that the fine-tuned Flan-T5 excels in moral evaluations.

**Strengths:**

- The proposed benchmark consists of a series of text-based games that evaluate the moral reasoning of language models through conversations. The game environment incorporates semantic matching, allowing for better alignment of the language model's output to valid actions.
- The games were designed with a focus on the moral dimension, a crucial yet under-researched metric for language models. The definitions and considerations regarding moral principles in these games are robust, drawing from established booklists or datasets.
- The authors conducted comprehensive experiments to assess the moral capabilities of leading open-source models. The evaluation was performed on both the proposed benchmark and other moral corpora.

**Weaknesses:**

About the Proposed Benchmark:
- The authors mention that "Each game is generally five to eight nodes deep, has between ten to twenty unique conversation states, and about ten unique triggers connecting these nodes."  Such a game's state space seems relatively basic to challenge the multi-step reasoning of language models. Perhaps for some leading commercial LLMs, this benchmark with a 5-8 level depth might be too simplistic. While the authors attribute this simplicity to the context window constraints of language models, advances in the field suggest there are methods to overcome this limitation. Introducing games of varied complexity, especially focusing on the number of dialogue rounds, could provide a more rigorous test for the moral reasoning of advanced models.
- The game's moral scores are manually annotated. It is important to detail the selection and background of the annotators. Are they trained? How is their annotation accuracy evaluated, and are their annotation cross-validated?

About the Experiments:
- The explanation about the loss score (section 7.1) lacks clarity. Could you elucidate how the moral scores were normalized? It's a bit puzzling to see game scores ranging from -100 to 100 being mapped to -3 to 2. The authors mention that "These scores were normalized and weighted so negative scores, −3 ≤ norm_score ≤ 0, were more impactful than positive ones, 0 ≤ norm_score ≤ 2." raises some questions. For the sake of model improvement, wouldn't it generally be beneficial to give more emphasis to positive samples during the fine-tuning process, leading to more significant weight adjustments?
- The model was trained and evaluated on the proposed benchmark. This improved its performance on that dataset, but doesn't necessarily demonstrate its ability to generalize well. The authors also conducted experiments on other datasets. However, what is the distribution between these two datasets? Do they share similarities?
- How was the temperature parameter of the model set in the experiments? Would different settings affect the evaluation results?
- Would the input order of valid actions have any impact on the evaluation results? I seem to remember a study suggesting that the decisions of GPT-3.5 and GPT-4 might be influenced by the sequence of the options given. Have you found similar observations?

About the Writing:

This paper might benefit from some adjustments:
- Incorrect citation format: cite/citep.
- The section on related work could delve deeper into topics like text-based game benchmarks, LM for gameplay, and morality in LMs.

**Questions:**

Two minor questions:
- In Section 3, the title is quite broadly. It might be helpful to specify the specific games, such as "Moral High Ground". I noticed that the initial concepts of 'live games' and 'control' aren't revisited later on. Instead, terms like BCG games and social chemistry games are introduced. It would be great to understand their correlation.
- How was the metric Win Rate determined?

---

> ### Author Response · Authors · 2023-11-16
>
> Thank you for your comments. We have made the following revisions to address your concerns. We format our changes in a similar ordering to yours so finding the changes that directly address your concerns should be easier.
>
> * Section 3.1: added a brief justification for our usage of smaller game sizes.
>
>   * “Our goal with this structure was to make the game complex enough to be non-trivial yet simple enough to be navigated by language models while keeping their context windows small. This also facilitated quicker manual game creation. In our early testing, larger games or games with many choices per node yielded poor model performance since, even though the prompts did not fill the context windows, they were still long enough to confuse the models“
>
> * Section 5.1: added more information about who the annotators were and measures to reduce bias.
>
>   * “These games were created and annotated manually by the authors with response scores.“
>
>   * “Discussion between authors was held on scores to avoid significant bias.“
>
> * Section 7.1: added clarification as to how the loss scores were normalized and weighted.
>
>   * “First, these scores were normalized from [-100, 100] to [-1, 1]. Next, each score was also given one of two weights, one for positive scores and another for negative scores. We tried several different weights, but weighting negative scores by 3, $-3 \le norm\\_score < 0$, and positive scores by 2, $0 \le norm\\_score \le 2$, yielded the best learning performance. This punishes the model for errors more harshly than a non-weighted loss. These scores were then used to create the final, biased loss.“
>
> * Section 5.3: added a footnote to clarify the model temperature and sampling method.
>
>   * “For, inference during all tests, the models were set with a standard temperature of 0.7 and a sampling decoding method to allow models to generate different responses after generating an invalid response.“
>
> * Section 5.1: added a clarification about how the valid actions are ordered when displayed.
>
>   * “The most notable of these is the "help" command. This command is used to display all possible responses to the current state of the conversation in an order that is randomized when each game is built.“
>
>   * Randomization at build time allows for the elimination of significant human bias in the ordering while allowing for consistency at runtime.
>
> * Changed \cite to \citep or \citet where appropriate.
>
> * Section 2: reworked the related works section to include more relevant papers and references.
>
> * Section 3: Changed the title to be more descriptive.
>
>   * “The Composition of Moral High Ground“
>
> * Section 3.2: added clarification as to how games are categorized between SCM, BCG, live, and control.
>
>   * “The following games are broken into two sets of two categories. First, the games are split between Social Chemistry games and BCG games. Second, within both of these two sets, those marked with ∗ are known as live games. These games present the model with a situation where some value is being violated or has the opportunity to be violated. Examples of these games are tbg-corrupt-1. The other games, marked with †, are the control. These games are designed to be very close in theme and structure to their live counterparts with mundane, moral conversations. Examples include tbg-anti-corrupt-1. Failing a live game means that a model gave into the immoral speaker and failing a control game indicates that the agent sought out an immoral path.“
>
> * Section 6: added information as to how win rate was calculated.
>
>   * “Here, Win Rate measures the percentage of games where the player obtained a positive score.“

---

### Official Review · Reviewer_Uvtk · 2023-11-03

**Soundness:** 1 poor
**Presentation:** 1 poor
**Contribution:** 1 poor
**Rating:** 1
**Confidence:** 4

**Summary:**

This paper introduces a text based gaming framework for evaluating LLMs for ethical understanding about the human society. Authors develop Moral High Ground game based on text-based games where an agent encounters various ethical situations while playing the game and it accumulates (positive/negative) points while taking actions in those situations. Each game is modeled as a conversation, where each utterance corresponds to a state (node in a graph) and response (action, an edge in the graph) leads to a new state. The final score in the game is indicative of the moral knowledge of the agent. Authors propose LLM as an agent playing the game to test their ethical knowledge.

**Strengths:**

1. The paper is addressing an important problem pertaining to ethical understanding of LLMs.
2. Authors create a framework for evaluating and comparing existing LLMs regarding moral knowledge.
3. Authors perform a detailed set of experiments to evaluate various LLMs and show that existing models lack understanding of ethical knowledge.

**Weaknesses:**

1. The paper is poorly written and has missing references (see below). The formatting of citations is not proper. There are grammatical errors (e.g., Section 5.1 "First, we created the games themselves").
2. The proposed framework doesn't involve any type of feedback or interaction between the agent and the environment. The agent provides a series of responses and accumulates score. This is unlike text-based game where an agent interacts and gets rewards and has the opportunity to improve. Authors should not call the current framework as text-based game (this is misleading) and distinguish their work from existing text-based games literature (references below).
3. The paper has limited novelty as authors are merely creating a new dataset of moral conversations based on existing corpora and use that to evaluate LLMs.

Missing references (these are not weaknesses per se):

1. There has been lot of work where LLMs have been evaluated for ethical and social norms, authors should discuss those in the related work, e.g.,

 * NormBank: A knowledge bank of situational social norms, Caleb Ziems, et al., ACL 2023
  * EtiCor: Corpus for Analyzing LLMs for Etiquettes, Ashutosh Dwivedi, et al., EMNLP 2023
  * Can Machines Learn Morality? The Delphi Experiment, Liwei Jiang, et al.

2. There has been lot of work on text-based games + LLMs. Various environments, agents and approaches have been proposed, authors should contrast and discuss some of these, e.g.,

* Text-based RL Agents with Commonsense Knowledge: New Challenges, Environments and Baselines., Keerthiram Murugesan et al., AAAI 20
*  ScriptWorld: Text Based Environment for Learning Procedural Knowledge, Abhinav Joshi, et al., IJCAI, 2023
*  Interactive Language Learning by Question Answering, Xingdi Yuan, et al., EMNLP 2019

**Questions:**

It is not clear how exactly the conversational graph is created from SocialChem dataset.

---

> ### Author Response · Authors · 2023-11-16
>
> Thank you for your comments.  We have made several revisions to address your concerns.  We have ordered them similarly to your original ordering and have cited specific places where we changed the text.
>
> * Fixed grammatical errors throughout the paper.
>
> * More appropriate appendix formatting.
>
> * Changed \cite to \citep or \citet where appropriate.
>
> * Section 3.1: added clarification about why we do not show the feedback to the models during play.
>
>   * “The score of each response is not shown to the model during gameplay to avoid influencing its responses. An important property of this benchmark is that it does not directly influence model responses; it measures the moral reasoning of agents as-is.“
>
> * Clarification: We utilize the term “game“ to describe the samples within our dataset as they are interactive environments in which agents can make choices that lead to different outcomes.  The primary difference between our games and others, e.g. Zork, is that feedback is not immediately given to the agent, rather the full score is calculated at the end.
>
> * Section 2: rewrote the related work section to include more appropriate papers such as Liwei Jiang, et al., Keerthiram Murugesan et al, and others not listed in the review.
>
> * Section 3.2.2: added clarification on how games were created from the social chemistry dataset.
>
>   * “These stories are not necessarily based on the situations within this dataset; they are created manually by using the axes as inspiration for the story’s topic.“

---

> > ### Comment · Reviewer_Uvtk · 2023-11-23
> >
> > I have read the comments by the authors. However, the response by authors does not address the fundamental issues with the work. In light of this, I will maintain my scores.